# Neural Distance Embeddings for Biological Sequences

**Gabriele Corso**[*]
MIT

**Rex Ying**
Stanford University

**Michal Pándy**
University of Cambridge

**Petar Veličković**
DeepMind

**Jure Leskovec**
Stanford University

**Pietro Liò**
University of Cambridge

## Abstract

The development of data-dependent heuristics and representations for biological sequences that reflect their evolutionary distance is critical for large-scale biological research. However, popular machine learning approaches, based on continuous Euclidean spaces, have struggled with the discrete combinatorial formulation of the edit distance that models evolution and the hierarchical relationship that characterises real-world datasets. We present Neural Distance Embeddings (`NeuroSEED`), a general framework to embed sequences in geometric vector spaces, and illustrate the effectiveness of the hyperbolic space that captures the hierarchical structure and provides an average 22% reduction in embedding RMSE against the best competing geometry. The capacity of the framework and the significance of these improvements are then demonstrated devising supervised and unsupervised `NeuroSEED` approaches to multiple core tasks in bioinformatics. Benchmarked with common baselines, the proposed approaches display significant accuracy and/or runtime improvements on real-world datasets. As an example for hierarchical clustering, the proposed pretrained and from-scratch methods match the quality of competing baselines with 30x and 15x runtime reduction, respectively.

## 1 Introduction

Over the course of evolution, biological sequences constantly mutate and a large part of biological research is based on the analysis of these mutations. Biologists have developed accurate statistical models to estimate the evolutionary distance between pairs of sequences based on their edit distance $D(s_1, s_2)$: the minimum number of (weighted) insertions, deletions or substitutions required to transform a string $s_1$ into another string $s_2$.

However, the computation of this edit distance kernel $D$ with traditional methods is bound to a quadratic complexity and hardly parallelizable, making its computation a bottleneck in large scale analyses, such as microbiome studies [1, 2, 3]. Furthermore, the accurate computation of similarities among multiple sequences, at the foundation of critical tasks such as hierarchical clustering and multiple sequence alignment, is computationally intractable even for relatively small numbers of sequences. Problems that in other spaces are relatively simple become combinatorially hard in the space of sequences defined by the edit distance. For example, finding the Steiner string, a classical problem in bioinformatics that can be thought of as computing the geometric median in the space of sequences, is NP-complete.

Classical algorithms and heuristics [4, 5, 6, 7] widely used in bioinformatics for these tasks are data-independent and, therefore, cannot exploit the low-dimensional manifold assumption that characterises real-world data [8, 9, 10]. Leveraging the available data to produce efficient and data-

---

[*]Correspondence to `gcorso@mit.edu`

35th Conference on Neural Information Processing Systems (NeurIPS 2021).

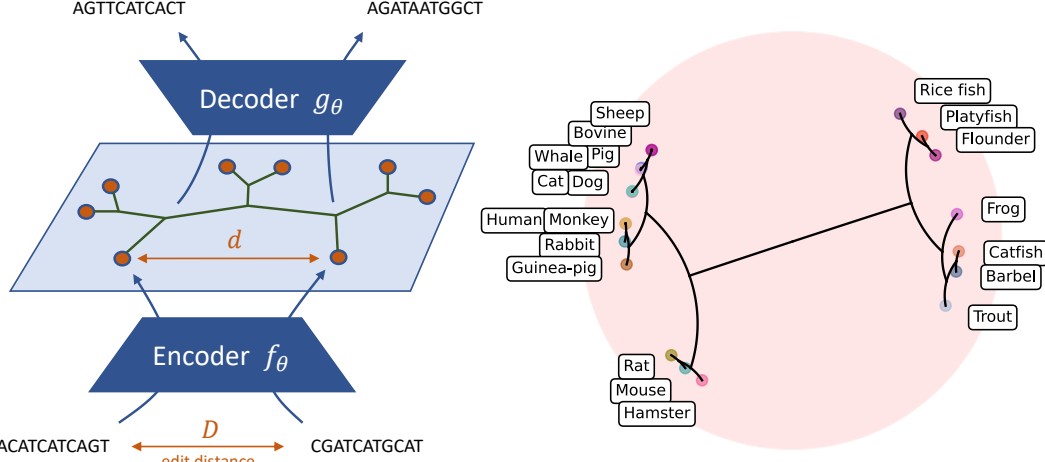

Figure 1: On the left, the key idea of `NeuroSEED`: learn an encoder function $f_\theta$ that preserves distances between the sequence and vector space ($D$ and $d$). The vector space can then be used to study the relationship between sequences and, potentially, decode new ones (see Section 7.2). On the right, an example of the *hierarchical clustering* produced on the Poincaré disk. The data was downloaded from UniProt [14] and consists of the P53 tumour protein from 20 different organisms.

dependent heuristics and representations would greatly accelerate large-scale analyses that are critical to biological research.

While the number of available biological sequences has grown exponentially over the past decades, machine learning approaches to problems related to string matching [11, 12] have not been adopted widely in bioinformatics due to their limitation in accuracy and speed. In contrast to most tasks in computer vision and NLP, string matching problems are typically formalised as combinatorial optimisation problems. These discrete formulations do not fit well with the current deep learning approaches. Moreover, representation learning methods based on Euclidean spaces struggle to capture the hierarchical structure that characterises real-world biological datasets due to evolution. Finally, common self-supervised learning approaches, very successful in NLP, are less effective in the biological context where relations tend to be between sequences rather than between bases [13].

In this work, we present Neural Distance Embeddings (`NeuroSEED`), a general framework to produce representations for biological sequences where the distance in the embedding space is correlated with the evolutionary distance $D$ between sequences. `NeuroSEED` provides fast approximations of the distance kernel $D$, low-dimensional representations for biological sequences, tractable analysis of the relationship between multiple sequences in the embedding geometry and a way to decode novel sequences.

Firstly, we reformulate several existing approaches into `NeuroSEED` highlighting their contributions and limitations. Then, we examine the task of embedding sequences to preserve the edit distance that is the basis of the framework. This analysis reveals the importance of data-dependent approaches and of using a geometry that matches the underlying data distribution well. The hyperbolic space is able to capture the implicit hierarchical structure given by biological evolution and provides an average 22% reduction in embedding RMSE against the best competing geometry.

We show the potential of the framework and its wide applicability by analysing two fundamental tasks in bioinformatics involving the relations between multiple sequences: *hierarchical clustering* and *multiple sequence alignment*. For both tasks, unsupervised approaches using `NeuroSEED` encoders are able to match the accuracy of common heuristics while being orders of magnitude faster. For *hierarchical clustering*, we also explore a method based on the continuous relaxation of Dasgupta's cost in the hyperbolic space which provides a 15x runtime reduction at similar quality levels. Finally, for *multiple sequence alignment*, we devise an original approach based on variational autoencoders that matches the performance of competitive baselines while significantly reducing the runtime complexity.

As a summary our contributions are: (*i*) We introduce `NeuroSEED`, a general framework to map sequences in geometric vector spaces, and reformulate existing approaches into it. (*ii*) We show how the hyperbolic space can bring significant improvements to the data-dependent analysis of biological sequences. (*iii*) We propose several heuristic approaches to classical bioinformatics problems that can be constructed on top of `NeuroSEED` embeddings and provide significant running time reduction against classical baselines.

## 2 Bioinformatics tasks

The field of bioinformatics has developed a wide range of algorithms to tackle the classical problems that we explore. Here we present the tasks and briefly mention their motivation and some of the baselines we test. More details are provided in Appendix B.

**Edit distance approximation**    In this work, we always deal with the classical edit distance where the same weight is given to every string operation, but all the approaches developed can be applied to any distance function of choice (which is given as an oracle). For example, when using amino acid sequences, one of the different metric variants of the classical substitution matrices such as mPAM250 [15] would be a good choice. As baseline approximation heuristics, we take k-mer [5], which is the most commonly used alignment-free method and represents sequences by the frequency vector of subsequences of a certain length, and FFP [16], another alignment-free method which looks at the Jensen-Shannon divergence between distributions of k-mers.

**Hierarchical clustering (HC)**    Discovering the intrinsic hierarchical structure given by evolutionary history is a critical step of many biological analyses. Hierarchical clustering (HC) consists of, given a pairwise distance function, defining a tree with internal points corresponding to clusters and leaves to datapoints. Dasgupta's cost [17] measures how well the tree generated respects the similarities between datapoints. As baselines we consider classical agglomerative clustering algorithms (Single [18], Complete [19] and Average Linkage [6]) and the recent technique [20] that uses a continuous relaxation of Dasgupta's cost in the hyperbolic space.

**Multiple sequence alignment (MSA)**    Aligning three or more sequences is used for the identification of active and binding sites as well as conserved protein structures, but finding its optimal solution is NP-complete. A related task to MSA is the approximation of the Steiner string which minimises the sum of the distances (consensus error) to the sequences in a set.

**Datasets**    To evaluate the heuristics we chose three datasets containing different portions of the 16S rRNA gene, crucial in microbiome analysis [21], one of the most promising applications of our approach. The first, Qiita [21], contains more than 6M sequences of up to 152 bp that cover the V4 hyper-variable region. The second, RT988 [11], has only 6.7k publicly available sequences of length up to 465 bp covering the V3-V4 regions. Both datasets were generated by Illumina MiSeq [22] and contain sequences of approximately the same length. Qiita was collected from skin, saliva and faeces samples, while RT988 was from oral plaques. The third dataset is the Greengenes full-length 16S rRNA database [23], which contains more than 1M sequences of length between 1,111 to 2,368. Moreover, we used a dataset of synthetically generated sequences to test the importance of data-dependent approaches. A full description of the data splits for each of the tasks is provided in Appendix B.4.

## 3 Neural Distance Embeddings

The underlying idea behind the `NeuroSEED` framework, represented in Figure 1, is to map sequences in a continuous space so that the distance between embedded points is correlated to the one between sequences. Given a distribution of sequences and a distance function $D$ between them, any `NeuroSEED` approach is formed by four main components: an embedding geometry, an encoder model, a decoder model, and a loss function.

**Embedding geometry**    The distance function $d$ between the embedded points defines the geometry of the embedding space. While this factor has been mostly ignored by previous work [11, 24, 25, 26, 27], we show that it is critical for this geometry to reflect the relations between the sequences in the domain. In our experiments, we provide a comparison between Euclidean, Manhattan, cosine, squared Euclidean (referred to as Square) and hyperbolic distances (details in Appendix D).

**Encoder model** The encoder model $f_\theta$ maps sequences to points in the embedding space. In this work we test a variety of models as encoder functions: linear layer, MLP, CNN, GRU [28] and transformer [29] with local and global attention. The details on how the models are adapted to the sequences are provided in Appendix C. Chen *et al.* [24] proposed CSM, an encoder architecture based on the convolution of subsequences. However, as also noted by Koide *et al.* [12], this model does not perform well when various layers are stacked and, due to the interdependence of cells in the dynamic programming routine, it cannot be efficiently parallelised on GPU.

**Decoder model** For some tasks it is also useful to decode sequences from the embedding space. This idea, employed in Section 7.2 and novel among the works related to `NeuroSEED`, enables to apply the framework to a wider set of problems.

**Loss function** The simplest way to train a `NeuroSEED` encoder is to directly minimise the MSE between the sequences' distance and its approximation as the distance between the embeddings:

$$L(\theta, S) = \sum_{s_1, s_2 \in S} (D(s_1, s_2) - \alpha \, d(f_\theta(s_1), f_\theta(s_2)))^2 \tag{1}$$

where $\alpha$ is a constant or learnable scalar. Depending on the application that the learned embeddings are used for, the MSE loss may be combined or substituted with other loss functions such as the triplet loss for *closest string retrieval* (Appendix F), the relaxation of Dasgupta's cost for *hierarchical clustering* (Section 7.1) or the sequence reconstruction loss for *multiple sequence alignment* (Section 7.2).

There are at least five previous works [11, 24, 25, 26, 27] that have used approaches that can be described using the `NeuroSEED` framework. These methods, summarised in Table 1, show the potential of approaches based on the idea of `NeuroSEED`, but share two significant limitations. The first is the lack of analysis of the geometry of the embedding space, which we show to be critical. The second is that the range of tasks is limited to *edit distance approximation* (EDA) and *closest string retrieval* (CSR). We highlight how this framework has the flexibility to be adapted to significantly more complex tasks involving relations between multiple sequences such as *hierarchical clustering* and *multiple sequence alignment*.

Table 1: Summary of the previous and the proposed `NeuroSEED` approaches. EDA stands for edit distance approximation and CSR for closest string retrievals. For our experiments, in the columns geometry and encoder we report those that performed best among the ones tested.

| Method | Geometry | Encoder | Decoder | Loss | Tasks |
|--------|----------|---------|---------|------|-------|
| Zheng *et al.* [11] | Jaccard | CNN | ✗ | MSE | EDA |
| Chen *et al.* [24] | Cosine | CSM | ✗ | MSE | EDA |
| Zhang *et al.* [25] | Euclidean | GRU | ✗ | MAE + triplet | EDA & CSR |
| Dai *et al.* [26] | Euclidean | CNN | ✗ | MAE + triplet | EDA & CSR |
| Gomez *et al.* [27] | Square | CNN | ✗ | MSE | EDA & CSR |
| Section 5 | Hyperbolic | CNN & transformer | ✗ | MSE | EDA |
| Section 6 | Hyperbolic | CNN & transformer | ✗ | MSE | HC & MSA |
| Section 7.1 | Hyperbolic | Linear | ✗ | Relaxed Dasgupta | HC |
| Section 7.2 | Cosine | Linear | ✓ | MSE + reconstr. | MSA |
| Appendix F | Hyperbolic | CNN & transformer | ✗ | MSE & triplet | CSR |

## 4 Related work

**Hyperbolic embeddings** Hyperbolic geometry is a non-Euclidean geometry with constant negative sectional curvature and is often referred to as a continuous version of a tree for its ability to embed trees with arbitrarily low distortion. The advantages of mapping objects with implicit or explicit hierarchical structure in the hyperbolic space have also been shown in other domains [30, 31, 32, 10]. In comparison, this work deals with a very different space defined by the edit distance in biological sequences and, unlike most of the previous works, we do not only derive embeddings for a particular set of datapoints, but train an encoder to map arbitrary sequences from the domain in the space.

**Sequence Distance Embeddings**   The clear advantage of working in more computationally tractable spaces has motivated significant research in *Sequence Distance Embeddings* [33] (also known as *low-distortion embeddings*) approaches to variants of the edit distance [34, 35, 36, 37, 38, 39, 40]. However, they are all *data-independent* and have shown weak performance on the 'unconstrained' edit distance.

**Hashing and metric learning**   `NeuroSEED` also fits well into the wider research on *learning to hash* [41], where the goal is typically to map a high dimensional vector space into a smaller one preserving distances. Another field related to `NeuroSEED` is *metric learning* [42, 43], where models are trained to learn embeddings from examples of similar and dissimilar pairs.

**Locality-sensitive hashing**   Finally, the work presented is complementary to the line of research in locality-sensitive hashing methods. Researchers have developed a series of heuristics especially for the tasks of sequence clustering [44, 45] and local alignment [46]. These use as subroutines embeddings/features based on alignment-based or alignment-free methods, and therefore, fall into the limitations we highlight in the paper. Future work could analyse how to improve these heuristics with `NeuroSEED` embeddings.

## 5   Edit distance approximation

In this section we test[2] the performance of different encoder models and distance functions to preserve an approximation of the edit distance in the `NeuroSEED` framework trained with the MSE loss. To make the results more interpretable and comparable across datasets, we report results using % RMSE:

$$\% \,\mathrm{RMSE}(\theta, S) = \frac{100}{n} \sqrt{L(\theta, S)} = \frac{100}{n} \sqrt{\sum_{s_1, s_2 \in S} (ED(s_1, s_2) - n\, d(f_\theta(s_1), f_\theta(s_2)))^2} \quad (2)$$

where $n$ is the maximum sequence length. This can be interpreted as an approximate average error in the distance prediction as a percentage of the size of the sequences.

| Model | RT988 Baseline | RT988 Hyperbolic | Qiita Baseline | Qiita Hyperbolic | Greengenes Baseline | Greengenes Hyperbolic | Training/Inference |
|---|---|---|---|---|---|---|---|
| NW alignment | - | - | - | - | - | - | - / 17.5h |
| 4-mer | 1.79 | - | 6.01 | - | 5.93 | - | 7s / 7s |
| 5-mer | 1.41 | - | 5.03 | - | 3.60 | - | 29s / 29s |
| 6-mer | 1.47 | - | 5.72 | - | 3.15 | - | 118s / 118s |
| FFP 8 | 12.03 | - | 20.42 | - | 10.26 | - | 360s / 360s |
| FFP 9 | 11.86 | - | 17.53 | - | 8.63 | - | 679s / 679s |
| FFP 10 | 10.80 | - | 16.16 | - | 14.13 | - | 1274s / 1274s |
| Linear | 21.36±7.07 | 0.51±0.01 | 4.39±0.09 | 2.50±0.01 | 1155.74±18.34 | 2.70±0.01 | 1.1h / 3s |
| MLP | 1.10±0.05 | 0.59±0.20 | 4.36±0.19 | 1.85±0.02 | 4.38±0.13 | 2.53±0.03 | 0.9h / 3s |
| CNN | 0.58±0.05 | 0.59±0.01 | 2.68±0.05 | **1.56±0.01** | 1.37±0.04 | **1.00±0.01** | 2.1h / 6s |
| GRU | 1.10±0.11 | 2.56±3.33 | 3.30±0.06 | 2.60±0.16 | 1.61±0.02 | 1.18±0.16 | 7.4h / 65s |
| Global T. | 0.52±0.01 | **0.46±0.01** | 2.10±0.05 | 1.83±0.03 | 2.09±0.03 | 1.91±0.07 | 2.2h / 3s |
| Local T. | 0.57±0.00 | **0.45±0.01** | 2.42±0.02 | 1.86±0.02 | 1.85±0.04 | 1.89±0.05 | 2.0h / 3s |

Figure 2: % RMSE test set results (4 runs). All neural models have an embedding space dimension of 128. The baseline geometries were chosen by best average performance and correspond to cosine distance for k-mers, Jensen-Shannon divergence for FFP and Euclidean distance for the neural models. The baselines CNN and GRU correspond, therefore, with [26] and [25]. The training and inference time comparisons provided in the Greengenes dataset are shown for a set of 5k sequences and were all run on a CPU (Intel Core i7) with the exception of the neural models' training that was run on GPU (GeForce GTX TITAN Xp). The distance function used does not significantly impact the runtime of any of the models. In all the tables: T. stands for transformer, - indicates that is not applicable or the model did not converge, **bold** the best results and the green-to-white colour scale the range of results best-to-worst. Full results for every geometry can be found in Figure 10 in Appendix E.

---

[2]Code, datasets and tuned hyperparameters can be found at https://github.com/gcorso/NeuroSEED.

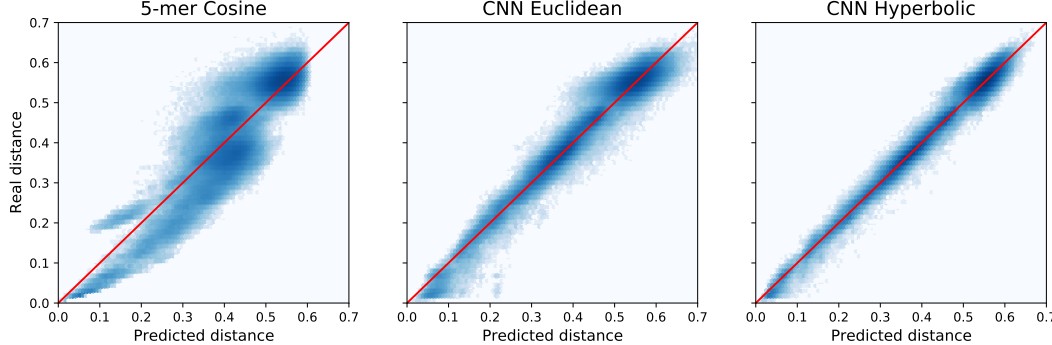

Figure 3: Qualitative comparison in the Qiita dataset between the best performing baseline (5-mer with cosine distance) and the CNN in the Euclidean and hyperbolic space. For every test set sequence pair, predicted vs real distances are plotted, the darkness represents the density of points. The CNN model follows much more tightly the red line of the oracle across the whole range of distances in the hyperbolic space.

**Data-dependent vs data-independent methods** Figures 2 and 3 show that, across the datasets and the distance functions, the data-dependent models learn significantly better representations than data-independent baselines. The main reason for this is their ability to focus on and dedicate the embedding space to a manifold of much lower dimensionality than the complete string space. This observation is further supported by the results in Appendix E, where the same models are trained on synthetic random sequences and the data-independent baselines are able to better generalise.

Our analysis also confirms the results from Zheng *et al.* [11] and Dai *et al.* [26] which showed that convolutional models outperform feedforward and recurrent models. We also show that transformers, even when with local attention, produce, in many cases, better representations. Attention could provide significant advantages when considering more complex definitions of distance that include, for example, inversions [47], duplications and transpositions [48].

**Hyperbolic space** The most interesting and novel results come from the analysis of the geometry of the embedding space. In these biological datasets, there is an implicit hierarchical structure that is well reflected by the hyperbolic space. Thanks to this close correspondence even relatively simple models perform very well with the hyperbolic distance. In convolutional models, the hyperbolic space provides a 22% average RMSE reduction against the best competing geometry for each model.

The benefit of using the hyperbolic space is clear when analysing the dimension required (Figure 4). The hyperbolic space provides significantly more efficient embeddings, with the model reaching the 'elbow' at dimension 32 and matching the performance of the other spaces with dimension 128 with only 4 to 16. Given that the space to store the embeddings and the time to compute distances between them scale linearly with the dimension, this provides a significant improvement in downstream tasks over other `NeuroSEED` approaches.

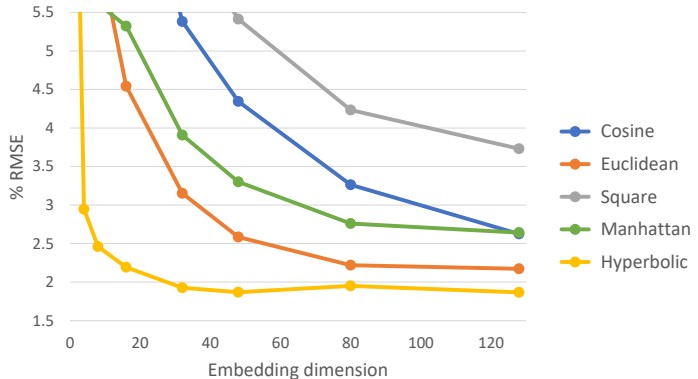

Figure 4: *Edit distance approximation* % RMSE on Qiita dataset for a global transformer with different distance functions.

**Running time**  The computational complexity of approximating the pairwise distance matrix[3] for $N$ sequences of length $M$ is reduced from $O(N^2 \ M^2/\log M)$ to $O(N(M+N))$ assuming constant embedding size and a model linear with respect to the sequence length. This translates into a very significant runtime improvement even when comparing only 5000 sequences, as can be seen from the last column in Figure 2. Moreover, while the training takes longer than the baselines due to the more complex optimisation, in applications such as microbiome analysis, biologists typically analyse data coming from the same distribution (e.g. the 16S rRNA gene) for multiple individuals, therefore the initial cost would be significantly amortised.

## 6  Unsupervised heuristics

In this section, we show how competitive heuristics for *hierarchical clustering* and *multiple sequence alignment* can be built on the low-distortion embeddings produced by the models trained in the previous section.

**Hierarchical clustering**  Agglomerative clustering, the most commonly used class of HC algorithms, can be accelerated when run directly on `NeuroSEED` embeddings produced by the pretrained model. This reduces the complexity to generate the pairwise distance matrix from $O(N^2M^2/\log M)$ to $O(N(M+N))$ and allows to accelerate the clustering itself using geometric optimisations like locality-sensitive hashing.

We test models with no further tuning from Section 5 on a dataset of 10k unseen sequences from the Qiita dataset. The results (Figure 5) show that there is no statistical difference in the quality of the hierarchical clustering produced with ground truth distances compared to that with convolutional or attention hyperbolic `NeuroSEED` embeddings. Instead, the difference in Dasgupta's cost between different architectures and geometries is statistically significant and it results in a large performance gap when these trees are used for tasks such as MSA shown below. The total CPU time taken to construct the tree is reduced from more than 30 minutes to less than one in this dataset and the difference gets significantly larger when scaling to datasets of more and longer sequences.

| Baselines | | Model | Cosine | Euclidean | Square | Manhattan | Hyperbolic |
|---|---|---|---|---|---|---|---|
| | | **4-mer** | 0.261 | 0.260 | 0.242 | 0.191 | 0.299 |
| **Single L.** | 0.628 | **Linear** | $0.062_{\pm0.007}$ | $0.172_{\pm0.036}$ | $0.153_{\pm0.037}$ | $0.177_{\pm0.026}$ | $0.028_{\pm0.005}$ |
| **Complete L.** | 0.479 | **MLP** | $0.169_{\pm0.054}$ | $0.095_{\pm0.021}$ | $0.289_{\pm0.094}$ | $0.178_{\pm0.029}$ | $0.035_{\pm0.004}$ |
| **Average L.** | **0.000** | **CNN** | $0.028_{\pm0.003}$ | $0.030_{\pm0.004}$ | $0.067_{\pm0.022}$ | $0.081_{\pm0.047}$ | $\mathbf{-0.004_{\pm0.015}}$ |
| | | **GRU** | - | $0.042_{\pm0.006}$ | $0.068_{\pm0.010}$ | $0.069_{\pm0.015}$ | $0.066_{\pm0.043}$ |
| | | **Global T.** | $0.032_{\pm0.014}$ | $\mathbf{0.003_{\pm0.008}}$ | $0.038_{\pm0.005}$ | $\mathbf{0.002_{\pm0.003}}$ | $\mathbf{0.000_{\pm0.006}}$ |
| | | **Local T.** | $0.035_{\pm0.003}$ | $0.022_{\pm0.008}$ | $0.034_{\pm0.005}$ | $0.022_{\pm0.003}$ | $\mathbf{0.000_{\pm0.007}}$ |

Figure 5: Average Linkage % increase in Dasgupta's cost of `NeuroSEED` models compared to the performance of clustering on the ground truth distances, ubiquitously used in bioinformatics. Average Linkage was the best performing clustering heuristic across all models.

**Multiple sequence alignment**  Clustal, the most popular MSA heuristic, is formed by a phylogenetic tree estimation phase that produces a guide tree then used by a progressive alignment phase to compute the complete alignment. However, the first of the two phases, based on hierarchical clustering, is typically the bottleneck of the algorithm. On 1200 RT988 sequences (used below), the construction of the guide tree takes 35 minutes compared to 24s for the progressive alignment. Therefore, it can be significantly accelerated using `NeuroSEED` heuristics to generate matrix and guide tree. In these experiments, we construct the tree running the Neighbour Joining algorithm (NJ) [49] on the `NeuroSEED` embeddings and then pass it on the Clustal command-line routine that performs the alignment and returns an alignment score.

Again, the results reported in Figure 6 show that the alignment scores obtained when using the `NeuroSEED` heuristics with attention models are not statistically different from those obtained with

---

[3]Computing the pairwise distance matrix of a set of sequences is a critical step of many algorithms including the ones analysed in the next section.

the ground truth distances. Most of the models also show a relatively large variance in performance across different runs. This has positive and negative consequences: the alignment obtained using a single run may not be very accurate, but, by training an ensemble of models and applying each of them, we are likely to obtain a significantly better alignment than the baseline while still only taking a fraction of the time.

| Model | Cosine | Euclidean | Hyperbolic |
|---|---|---|---|
| **Linear** | 60.6±35.1 | 111.3±3.6 | 57.5±22.0 |
| **MLP** | 72.3±11.8 | 53.6±3.1 | -11.7±18.9 |
| **CNN** | 31.0±16.2 | 4.7±9.7 | **-16.3±16.1** |
| **Global T.** | 39.4±74.3 | 1.9±3.8 | 31.1±21.8 |
| **Local T.** | 31.9±30.5 | 8.6±14.1 | **-20.1±7.3** |

Figure 6: Percentage improvement (average of 3 runs) in the alignment cost (the lower the better) returned by Clustal when using the heuristics to generate the tree as opposed to its default setting using NJ on real distances.

# 7 Supervised heuristics

In this section we propose and evaluate two methods to adapt `NeuroSEED` to the tasks of *hierarchical clustering* and *multiple sequence alignment* with tailored loss functions.

## 7.1 Relaxed approach to hierarchical clustering

An alternative approach to *hierarchical clustering* uses the continuous relaxation [20] of Dasgupta's cost [17] as loss function to embed sequences in the hyperbolic space (for more details see Appendix B.2 and [20]). In comparison to Chami *et al.* [20], we show that it is possible to significantly decrease the number of pairwise distances required by directly mapping the sequences into the space. This allows to considerably accelerate the construction, especially when dealing with a large number of sequences without requiring any pretrained model. Figure 1 shows an example of the relaxed approach when applied to a small dataset of proteins.

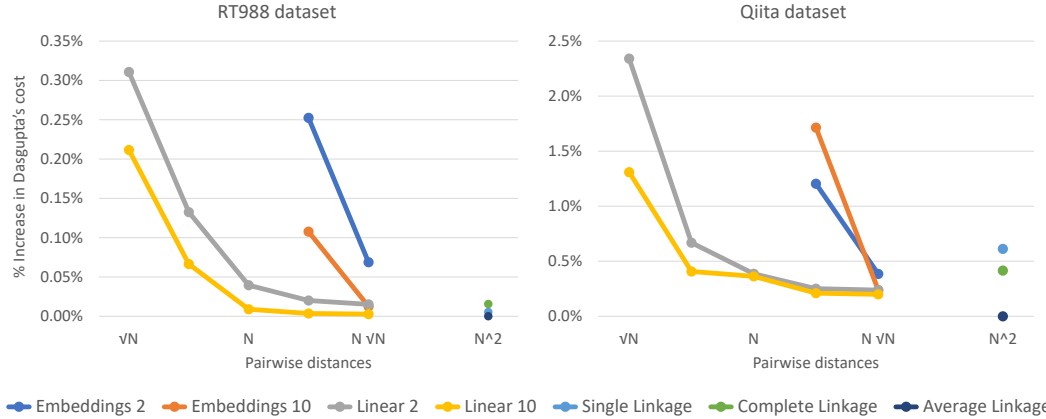

Figure 7: Average Dasgupta's cost of the various approaches with respect to the number of pairwise distances used in the RT988 and Qiita datasets. The performances are reported as the percentage increase in cost compared to the one of the Average Linkage (best performing). Embedding refers to the baseline [20] while Linear to the relaxed `NeuroSEED` approach. The attached number represents the dimension of the hyperbolic space used.

The results, plotted in Figure 7, show that a simple linear layer mapping sequences to the hyperbolic space is capable of obtaining with only $N$ pairwise distances very similar results to those from

agglomerative clustering ($N^2$ distances) and hyperbolic embedding baselines ($N\sqrt{N}$ distances). In the RT988 dataset this corresponds to, respectively, 6700x and 82x fewer labels and leads to a reduction of the total running time from several hours (>2.5h on CPU for agglomerative clustering, 1-4h on GPU for hyperbolic embeddings) to less than 10 minutes on CPU for the relaxed `NeuroSEED` approach (including label generation, training and tree inference) with no pretraining required. Finally, using more complex encoding architectures such as MLPs or CNNs does not provide significant improvements.

## 7.2   Steiner string approach to multiple sequence alignment

An alternative approach to *multiple sequence alignment* uses a decoder from the vector space to convert the Steiner string approximation problem (Appendix B.3) in a continuous optimisation task.

This method, summarised in Figure 8 and detailed in Appendix G, consists of training an autoencoder to map sequences to and from a continuous space preserving distances using only pairs of sequence at a time (where calculating the distance is feasible). This is achieved by combining in the loss function a component for the latent space edit distance approximation and one for the reconstruction accuracy of the decoder. The first is expressed as the MSE between the edit distance and the vector distance in the latent space. The second consists of the mean element-wise cross-entropy loss of the decoder's outputs with the real sequences. At test time the encoder embeds all the sequences in the set, the geometric median point (minimising the sum of distances in the embedding space) is found with a relatively simple optimisation procedure and then the decoder is used to find an approximation of the Steiner string. During training, Gaussian noise is added to the embedded point in the latent space forcing the decoder to be robust to points not directly produced by the encoder.

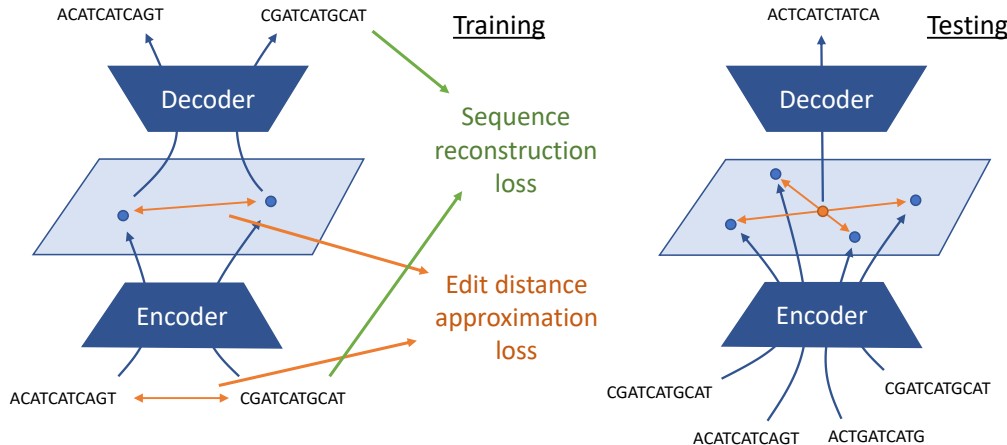

Figure 8: Diagram for the Steiner string approach to *multiple sequence alignment*. On the left, the training procedure using pairs of sequences and a loss combining edit distance approximation and sequence reconstruction. On the right the extrapolation for the generation of the Steiner string by decoding the geometric median in the embedding space.

As baselines, we report the average consensus error (average distance to the strings in the set) obtained using: a random sequence in the set (random), the centre string of the set (centre) and two competitive greedy heuristics (greedy-1 and greedy-2) proposed respectively by [50] and [51].

| Baselines | |
|---|---|
| Random | 75.98 |
| Centre | 62.52 |
| Greedy-1 | **59.43** |
| Greedy-2 | **59.41** |

| Model | Cosine | Euclidean | Square | Hyperbolic |
|---|---|---|---|---|
| Linear | **59.41±0.11** | 59.96±0.27 | 60.53±0.49 | 60.89±0.82 |
| MLP | 60.80±0.35 | 60.00±0.18 | 59.81±0.22 | 59.86±0.12 |
| CNN | 60.96±0.48 | 60.20±0.26 | 60.76±1.09 | 60.48±0.52 |

Figure 9: Average consensus error for the baselines (left) and `NeuroSEED` models (right).

The results show that the models consistently outperform the centre string baseline and are close to the performance of the greedy heuristics suggesting that they are effectively decoding useful information from the embedding space. The computational complexity for $N$ strings of size $M$ is reduced from $O(N^2M^2/\log M)$ for the centre string and $O(N^2M)$ for the greedy baselines to $O(NM)$ for the proposed method. Future work could employ models that directly operate in the hyperbolic space [52] to further improve the performance.

# 8    Conclusion

In this work, we proposed and explored Neural Distance Embeddings, a framework that exploits the recent advances in representation learning to embed biological sequences in geometric vector spaces. By studying the capacity to approximate the evolutionary edit distance between sequences, we showed the strong advantage provided by the *hyperbolic space* which reflects the biological hierarchical structure.

We then demonstrated the effectiveness and wide applicability of `NeuroSEED` on the problems of *hierarchical clustering* and *multiple sequence alignment*. For each task, we experimented with two different approaches: one unsupervised tying `NeuroSEED` embeddings into existing heuristics and a second based on direct exploitation of the geometry of the embedding space via a tailored loss function. In all cases, the proposed approach performed on par with or better than existing baselines while being significantly faster.

Finally, `NeuroSEED` provides representations that are well suited for human interaction as the embeddings produced can be visualised and easily interpreted. Towards this goal, the very compact representation of hyperbolic spaces is of critical importance [10]. This work also opens many opportunities for future research direction with different types of sequences, labels, architectures and tasks. We present and motivate these directions in Appendix A.

## Acknowledgements

The authors thank Saro Passaro for his insightful comments and suggestions and Frank Stajano, Andrei Margeloiu, Hannes Stärk and Cristian Bodnar for their review of the manuscript.

## Disclosure of Funding

G.C. is funded by the Robert Shillman (1974) Fellowship at MIT. P.V. is a Research Scientist at DeepMind. J.L. is a Chan Zuckerberg Biohub investigator. We also gratefully acknowledge the support of DARPA under Nos. HR00112190039 (TAMI), N660011924033 (MCS); ARO under Nos. W911NF-16-1-0342 (MURI), W911NF-16-1-0171 (DURIP); NSF under Nos. OAC-1835598 (CINES), OAC-1934578 (HDR), CCF-1918940 (Expeditions), IIS-2030477 (RAPID), NIH under No. R56LM013365; Stanford Data Science Initiative, Wu Tsai Neurosciences Institute, Chan Zuckerberg Biohub, Amazon, JPMorgan Chase, Docomo, Hitachi, Intel, JD.com, KDDI, NVIDIA, Dell, Toshiba, Visa, and UnitedHealth Group.

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
