# A Limitations and future work

We believe that the `NeuroSEED` framework has the potential to be applied to numerous problems and this work constitutes an initial analysis of its geometrical properties and applications. Here, we list some of the limitations of the current analysis and the potential directions of research to cover them.

**Type of sequences**    All real-world datasets analysed consist of sequence reads of the same part of the genome. This is a widespread set-up for sequence analysis but not ubiquitous. Shotgun metagenomics consists of sequencing random parts of the genome. This would generate sequences lying on a low-dimensional manifold where the hierarchical relationship of evolution is combined with the relationship based on the specific position in the whole genome. Therefore, more complex geometries, such as product spaces [53, 54], might be best suited.

**Type of labels**    In this project, we work with edit distances between sequences, these are too expensive for large-scale analysis, but it is feasible to produce a large enough training set. For different definitions of distance, however, this might not be the case, future work could explore the robustness of this framework to inexact estimates of the distances as labels.

**Architectures**    Throughout the project, we used models that have been shown to work well for other types of sequences and tasks. However, the correct inductive biases that models should have to perform `NeuroSEED` might be different and even dependent on the type of distance they try to preserve. [24, 12] provide some initial work in this direction with respect to the edit distance. Moreover, the capacity of the hyperbolic space could be further exploited using models that directly operate in the space [52, 55, 56].

**Self-supervised embeddings**    Finally, the direct use of the embeddings produced by `NeuroSEED` for downstream tasks would enable the application of a wide range of geometric data processing tools to the analysis of biological sequences.

**Long-term impact**    We believe the combination of `NeuroSEED` embeddings and geometric deep learning [57, 58] techniques could be beneficial to analyse and track the spectrum of mutations in a wide variety of biological and medical applications. This would have positive societal impacts in domains like microbiome analysis and managing epidemics. However, this could also have unethical applications in fields such as genome profiling.

# B Bioinformatics tasks

The field of bioinformatics has developed a wide range of algorithms to tackle the classical problems that we explore. We describe here the methods that are most closely related to our work. For a more comprehensive overview, the interested reader is recommended Gusfield [59] and Compeau *et al.* [60].

## B.1 Edit distance approximation

The task of finding the distance or similarity between two strings and the related task of global alignment lies at the foundation of bioinformatics.

**Alignment-based methods**    Classical algorithms to find the edit distance, such as Needleman–Wunsch [4], are based on the process of finding an alignment between the two strings via dynamic programming. However, these are bound to a quadratic complexity w.r.t. the length of the input sequence, the best algorithm [61] has a complexity $O(M^2/\log M)$ and there is evidence that this cannot be improved [62].

**Alignment-free methods**    With the rapid improvement of sequencing technologies and the subsequent increase in demand for large-scale sequence analyses, alternative computationally efficient sequence comparison methods have been developed under the category of alignment-free methods.

**k-mer** [5] is the most commonly used alignment-free method and basis for many other algorithms (such as FFP [16], ACS [63] and kmacs [64]). It considers all the sequences of a fixed length k, *k-mers*, and constructs a vector where each entry corresponds with the number of occurrences of a particular k-mer in the sequence. The distance between the strings is then approximated by some

type of distance $d$ between the vectors. Therefore, k-mer generates vectors of size $4^k$ and estimates the edit distance as $ED(s_1, s_2) \approx n\,\alpha\,d(\text{k-mer}(s_1), \text{k-mer}(s_2))$ where $\alpha$ is the only parameter of the model whose optimal value can be obtained with a single pass of the training set [4]:

$$\alpha^* = \arg\min_\alpha \sum_{ij} (r_{ij} - \alpha p_{ij})^2 \tag{3}$$

where $r_{ij} = n^{-1} ED(s_i, s_j)$ and $p_{ij} = d(\text{k-mer}(s_i), \text{k-mer}(s_j))$. Therefore:

$$\frac{\partial}{\partial \alpha} \sum_{ij} (r_{ij} - \alpha p_{ij})^2 |_{\alpha=\alpha^*} = 0$$

$$\sum_{ij} \frac{\partial}{\partial \alpha} (r_{ij}^2 - 2\alpha r_{ij} p_{ij} + \alpha^2 p_{ij}^2) |_{\alpha=\alpha^*} = 0 \tag{4}$$

$$\therefore \alpha^* = \frac{\sum_{ij} r_{ij} p_{ij}}{\sum_{ij} p_{ij}^2}$$

## B.2 Hierarchical clustering

**Single, Complete and Average Linkage**   The most common class of algorithms for *hierarchical clustering*, referred to as agglomerative methods, works in a bottom-up manner recursively merging similar clusters. These differ by the heuristics used to choose clusters to merge and include Single [18], Complete [19] and Average Linkage (or UPGMA) [6]. They typically run in $O(N^2 \log N)$ and require the whole $N^2$ distance matrix as input. Thus, with the edit distance, the total complexity is $O(N^2(M^2/\log M + \log N))$.

**Dasgupta's cost**   Dasgupta [17] proposed a global objective function that can be associated with the HC trees. Given a rooted binary tree $T$, for two datapoints $i$ and $j$ let $w_{ij}$ be their pairwise similarity, $i \vee j$ their lowest common ancestor in $T$ and $T[i \vee j]$ the subtree rooted at $i \vee j$. Dasgupta's cost of $T$ given $w$ is then defined as:

$$C_{\text{Dasgupta}}(T; w) = \sum_{ij} w_{ij}\,|\,\text{leaves}(T[i \vee j])\,| \tag{5}$$

In this work $w_{ij}$ is taken to be $1 - d_{ij}$ where $d_{ij}$ is the normalised distance between sequences $i$ and $j$.

## B.3 Multiple sequence alignment

*Multiple Sequence Alignment* (MSA) consists of aligning three or more sequences and is regularly used for phylogenetic tree estimation, secondary structure prediction and critical residue identification. Finding the global optimum alignment of $N$ sequences is NP-complete [65], therefore many heuristics have been proposed.

**Progressive alignment**   The most commonly used programs such as the Clustal series [7] and MUSCLE [66] are based on a phylogenetic tree estimation phase from the pairwise distances which produces a guide tree, which is then used to guide a progressive alignment phase. To replicate the classical edit distance used, Clustal is run with a substitution matrix with all the entries -1 except 0 on the main diagonal and gap opening and extension penalties equal to 1.

**Consensus error and Steiner string**   It is hard to quantify the goodness of a particular multiple alignment and there is no single well-accepted measure [59]. One option is to find the sequence $s^*$ that minimises the *consensus error* to the set of strings $S$: $E(s^*) = \sum_{s_i \in S} ED(s^*, s_i)$. The optimal string $s^*$ is known as *Steiner string*, while the *centre string* $s_c$ is the one in $S$ which minimises $E(s_c)$ and has an upper bound $E(s_c) \leq (2 - 2/M)E(s^*)$ [59]. Algorithms to find an approximation of the Steiner string typically use greedy heuristics [51, 50].

---

[4]Except when using the hyperbolic space, in which case the radius of the hypersphere to which points are projected and $\alpha$ are learned via gradient descent.

## B.4 Datasets

As real-world datasets, we used the Qiita, RT988 and Greengenes datasets of 16S rRNA subsequences. Experiments were also run on synthetic datasets formed by sequences randomly generated. In all datasets the splitting of sequences between train/val/test was random and duplicate sequences were discarded. Below we list the sizes of the datasets used for the results presented, these datasets can be downloaded from the public code repository.

**Edit distance approximation**   RT988 and Greengenes 5000/500/1200 sequences (train/val/test, 25M training pairwise distances), Qiita 7000/700/1500 sequences (49M distances), synthetic 70k/10k/20k sequences (3.5M distances).

**Hierarchical clustering**   the RT988 dataset is formed by 6.7k sequences to cluster while the Qiita one contains 10k sequences. The Qiita dataset used in the unsupervised approach is disjoint from the training set of the models.

**Multiple sequence alignment**   for the unsupervised approach the test set from the edit distance RT988 dataset was used, while the Steiner string approach was tested on the RT988 dataset using 4500/700 sequences for training/validation and 50 groups of 30 sequences for each of which the model computes an approximation of the Steiner string.

## C   Neural architectures

The framework of `NeuroSEED` is independent of the choice of architecture for the encoder. For each approach proposed in this project, we experiment with a series of models among the most commonly used in the literature for the analysis of sequences. In this section, we give some detail on how each model was adapted to the task at hand.

**Linear & MLP**   operate on the input sequence using the one-hot encodings, padding to the maximum sequence length and flattening as a vector.

**CNN**   is also applied to the padded sequence of one-hot elements. They are conceptually similar to the k-mer baseline with a few distinctions: CNNs can learn the kernels to apply, CNNs are equivariant not invariant to the translation of the patterns and, with multiple layers, CNNs can exploit hierarchical patterns in the data.

**GRU**   [28] operates on the sequence of one-hot sequence elements.

**Transformer**   [29] every token is formed by 4-16 bases and is given a specific positional encoding using sinusoidal functions. We test both global attention where every token queries all the others and local where it only queries its 2 neighbours. Local attention allows the model to have a complexity linear w.r.t. the number of tokens.

All the models are integrated with various forms of regularisation including weight decay, dropout [67], batch normalisation [68] and layer normalisation [69] and optimised using the Adam optimiser [70]. In the hyperbolic space, the embedded points are first projected on a hypersphere of learnable radius and then to the hyperbolic space.

## D   Distance functions

The key idea behind `NeuroSEED` is to map sequences into a vector space so that the distances in the sequence and the vector space are correlated. In this appendix, we present various definitions of distance in the vector space that we explored: L1 (referred as *Manhattan*), L2 (*Euclidean*), L2 squared (*square*), *cosine* and *hyperbolic* distances. For the hyperbolic space, we use the Poincaré ball model that embeds the points of the n-dimensional Riemannian manifold in an n-dimensional unit sphere $\mathbb{B}^n = \{x \in \mathbb{R}^n : \|x\| < 1\}$ where $\|\cdot\|$ denotes the Euclidean norm. Given a pair of vectors $\mathbf{p}$ and $\mathbf{q}$ of dimension $k$, the definitions for the distances are:

$$\text{Manhattan} \quad d(\mathbf{p}, \mathbf{q}) = \|\mathbf{p} - \mathbf{q}\|_1 = \sum_{i=0}^{k} |p_i - q_i| \tag{6}$$

$$\text{Euclidean} \quad d(\mathbf{p}, \mathbf{q}) = \|\mathbf{p} - \mathbf{q}\|_2 = \sqrt{\sum_{i=0}^{k} (p_i - q_i)^2} \tag{7}$$

$$\text{square} \quad d(\mathbf{p}, \mathbf{q}) = \|\mathbf{p} - \mathbf{q}\|_2^2 = \sum_{i=0}^{k} (p_i - q_i)^2 \tag{8}$$

$$\text{cosine} \quad d(\mathbf{p}, \mathbf{q}) = 1 - \frac{\mathbf{p} \cdot \mathbf{q}}{\|\mathbf{p}\| \|\mathbf{q}\|} = 1 - \frac{\sum_{i=0}^{k} p_i q_i}{\sqrt{\sum_{i=0}^{k} p_i^2} \sqrt{\sum_{i=0}^{k} q_i^2}} \tag{9}$$

$$\text{hyperbolic} \quad d(\mathbf{p}, \mathbf{q}) = \text{arcosh}\left(1 + 2\frac{\|\mathbf{p} - \mathbf{q}\|^2}{(1 - \|\mathbf{p}\|^2)(1 - \|\mathbf{q}\|^2)}\right) \tag{10}$$

## E Distortion on synthetic datasets

We used a dataset of randomly generated sequences to test the importance of data-dependent approaches and understand whether the improvements shown in Section 5 are brought by a better capacity of the neural models to model the edit distance mutation process or their ability to focus on the lower-dimensional manifold that the real-world data lies on.

| Model | RT988 | | | | | Qiita | | | | |
|---|---|---|---|---|---|---|---|---|---|---|
| | Cosine | Euclidean | Square | Manhattan | Hyperbolic | Cosine | Euclidean | Square | Manhattan | Hyperbolic |
| 2-mer | 7.782 | 4.927 | 8.000 | 5.036 | 4.859 | 21.222 | 11.752 | 30.453 | 11.639 | 10.481 |
| 3-mer | 3.392 | 3.351 | 3.520 | 2.987 | 3.308 | 12.352 | 7.962 | 32.219 | 7.439 | 6.657 |
| 4-mer | 1.790 | 3.314 | 1.899 | 2.318 | 3.294 | 6.006 | 7.015 | 34.098 | 5.636 | 6.728 |
| 5-mer | 1.409 | 3.449 | 1.422 | 1.801 | 3.470 | 5.027 | 7.638 | 34.559 | 5.391 | 7.600 |
| 6-mer | 1.471 | 3.710 | 1.450 | 1.686 | 3.730 | 5.723 | 8.383 | 34.616 | 5.844 | 8.275 |
| Linear | 0.62±0.03 | 21.3±7.0 | 27.2±10.8 | - | 0.51±0.01 | 3.38±0.06 | 4.39±0.09 | 5.83±0.21 | 3.82±0.09 | 2.50±0.01 |
| MLP | 1.57±0.16 | 1.10±0.05 | 6.78±2.50 | 1.01±0.04 | 0.59±0.20 | 4.98±0.11 | 4.36±0.19 | 8.52±0.78 | 4.92±0.10 | 1.85±0.02 |
| CNN | 0.69±0.03 | 0.58±0.05 | 2.95±1.09 | 0.98±0.06 | 0.59±0.01 | 2.54±0.04 | 2.68±0.05 | 5.03±0.85 | 4.06±0.21 | **1.56±0.01** |
| GRU | 14.9±4.56 | 1.10±0.11 | 1.96±0.47 | 1.13±0.15 | 2.56±3.33 | - | 3.30±0.06 | 5.52±0.15 | 3.74±0.01 | 2.60±0.16 |
| Global T. | 0.49±0.01 | 0.52±0.01 | 0.88±0.02 | **0.44±0.01** | **0.46±0.01** | 2.61±0.01 | 2.10±0.05 | 3.71±0.04 | 2.57±0.11 | 1.83±0.03 |
| Local T. | 0.51±0.03 | 0.57±0.00 | 0.58±0.02 | 0.48±0.01 | **0.45±0.01** | 2.67±0.04 | 2.42±0.02 | 3.72±0.06 | 2.46±0.02 | 1.86±0.02 |

| Model | Synthetic | | | | | Greengenes | | | | |
|---|---|---|---|---|---|---|---|---|---|---|
| | Cosine | Euclidean | Square | Manhattan | Hyperbolic | Cosine | Euclidean | Square | Manhattan | Hyperbolic |
| 2-mer | 10.49 | 7.11 | 10.53 | 7.28 | 7.11 | 16.172 | 7.983 | 14.753 | 7.931 | 5.084 |
| 3-mer | 5.71 | 6.02 | 5.81 | 6.01 | 5.99 | 11.210 | 5.583 | 10.994 | 5.352 | 5.133 |
| 4-mer | **3.74** | 6.24 | 3.87 | 5.92 | 6.23 | 5.931 | 3.874 | 5.981 | 3.611 | 5.164 |
| 5-mer | 3.92 | 6.75 | 3.97 | 5.72 | 6.75 | 3.600 | 3.427 | 3.339 | 3.107 | 5.182 |
| 6-mer | 4.71 | 7.26 | 4.72 | 5.37 | 7.31 | 3.152 | 3.478 | 2.828 | 2.905 | 5.192 |
| Linear | 4.77±0.04 | 33.9±35.1 | 5.25±0.03 | - | 6.50±0.60 | 3.60±0.05 | 1155±18.3 | 2670±3209 | 14133±680 | 2.70±0.01 |
| MLP | 9.79±0.08 | 9.40±0.05 | 7.74±0.05 | 9.82±0.06 | 10.71±0.18 | 4.60±0.08 | 4.38±0.13 | 8.73±0.77 | 3.97±0.06 | 2.53±0.03 |
| CNN | 4.18±0.25 | 4.93±0.04 | 4.93±0.03 | 5.48±0.06 | 4.60±0.15 | 1.83±0.05 | 1.37±0.04 | 2.23±0.03 | 1.58±0.03 | **1.00±0.01** |
| GRU | 6.30±4.93 | 5.11±0.10 | 5.60±4.33 | 5.68±0.22 | 8.54±0.84 | 24.69±0.00 | 1.61±0.02 | 24.69±0.00 | 4.90±0.69 | 1.18±0.16 |
| Global T. | 4.51±0.01 | 4.74±0.02 | 5.23±0.03 | 4.67±0.04 | 4.75±0.04 | 2.16±0.04 | 2.09±0.03 | 2.83±0.04 | 1.73±0.03 | 1.91±0.07 |
| Local T. | 4.45±0.03 | 4.86±0.03 | 5.05±0.03 | 4.87±0.02 | 4.49±0.03 | 2.12±0.02 | 1.85±0.04 | 2.37±0.05 | 1.72±0.06 | 1.89±0.05 |

Figure 10: % RMSE test set results on all datasets and for all models and distances for the edit distance approximation task. The embedding space dimensions are as in Figure 2.

In Figure 10, the picture that emerges from the results on the synthetic dataset is dramatically different from the one of real-world datasets and confirms the hypothesis that the advantage of neural models in real-world datasets is mainly due to their capacity to exploit the low-dimensional assumption.

Instead, in the synthetic dataset, the best neural models perform only on par (taking into account the difference embedding space dimension) with the baselines. This is caused by two related challenges: the incredibly large space of sequences ($4^{1024}$) that the model is trying to encode and the diversity between training and test sequences due to the random sampling. These make the task of learning a good encoding task too tough for currently feasible sizes of models and training data.

## F  Closest string retrieval

This task consists of finding the sequence that is closest to a given query among a large number of reference sequences and is very commonly used by biologists to classify newly sequenced genes.

**Task formulation**   Given a pretrained encoder $f_\theta$, its closest string prediction is taken to be the string $r_q \in R$ that minimises $d(f_\theta(r_q), f_\theta(q))$ for each $q \in Q$. This allows for sublinear retrieval via locality-sensitive hashing or other data structures which is critical in real-world applications where databases can have billions of reference sequences. As performance measures, we report the top-1, top-5 and top-10 percentage accuracies, where top-$k$ indicates the percentage of times the closest string is ranked in the top-$k$ predictions.

**Triplet loss**   The triplet loss [71, 72, 73] is widely used in the field of metric learning [42, 43] to learn embeddings that can be considered as a more direct form of supervision for this task. Given three examples with feature vectors $a$ (anchor), $p$ (positive) and $n$ (negative) where the $p$ is supposed to be closer to $a$ than $n$, the triplet loss is typically defined as:

$$L(a, p, n) = \max(0, d(a, p) - d(a, n) + m) \tag{11}$$

where $m$ is the safety margin and $d$ a given distance function between vectors (typically Euclidean or cosine).

| | Model | Cosine | | | Euclidean | | | Square | | | Manhattan | | | Hyperbolic | | |
|---|---|---|---|---|---|---|---|---|---|---|---|---|---|---|---|---|
| | | top 1 | top 5 | top 10 | top 1 | top 5 | top 10 | top 1 | top 5 | top 10 | top 1 | top 5 | top 10 | top 1 | top 5 | top 10 |
| **K-mer** | 2-mer | 25.5 | 42.4 | 50.8 | 23.0 | 40.7 | 49.2 | 23.0 | 40.7 | 49.2 | 21.5 | 38.6 | 47.3 | 25.5 | 42.4 | 50.8 |
| | 3-mer | 38.1 | 54.0 | 60.6 | 35.9 | 53.2 | 59.7 | 35.9 | 53.2 | 59.7 | 36.7 | 53.7 | 60.2 | 38.1 | 54.0 | 60.6 |
| | 4-mer | 43.8 | 60.3 | 66.9 | 41.5 | 58.3 | 64.3 | 41.5 | 58.3 | 64.3 | 43.2 | 59.4 | 65.8 | 43.8 | 60.3 | 66.9 |
| | 5-mer | 45.9 | 62.9 | 69.6 | 44.7 | 60.9 | 67.9 | 44.7 | 60.9 | 67.9 | 45.3 | 62.6 | 68.8 | 45.9 | 62.9 | 69.6 |
| | 6-mer | 45.5 | 62.7 | 68.2 | 44.9 | 60.9 | 67.3 | 44.9 | 60.9 | 67.3 | 44.9 | 62.6 | 68.3 | 45.5 | 62.7 | 68.2 |
| **MSE** | Linear | 47.7 | 65.1 | 72.2 | 38.6 | 49.9 | 54.1 | 42.5 | 54.1 | 58.8 | 39.8 | 50.3 | 53.8 | 43.2 | 63.7 | 71.4 |
| | MLP | 37.8 | 50.6 | 55.9 | 37.4 | 52.5 | 59.4 | 35.4 | 48.2 | 53.6 | 31.8 | 46.2 | 53.0 | 43.4 | 67.9 | 78.2 |
| | CNN | 47.0 | **75.5** | **84.2** | 40.0 | 65.3 | 75.2 | 38.1 | 62.4 | 72.3 | 32.0 | 52.9 | 62.2 | **50.1** | **77.2** | **85.9** |
| | GRU | - | - | - | 36.5 | 62.0 | 71.7 | 33.4 | 58.0 | 68.2 | 36.7 | 59.7 | 68.2 | 28.6 | 50.3 | 59.9 |
| | Global T. | **51.3** | 75.9 | **84.5** | 45.8 | 72.3 | 81.8 | 48.2 | 67.5 | 76.0 | 46.2 | 67.4 | 76.7 | **49.5** | 75.5 | 84.0 |
| | Local T. | **49.8** | 75.0 | **84.4** | 42.3 | 66.7 | 75.7 | 47.4 | 66.8 | 75.7 | 43.7 | 68.4 | 77.3 | 48.8 | 75.1 | **84.5** |
| **Triplet** | Linear | 47.4 | 70.1 | 78.2 | 41.4 | 53.6 | 58.6 | 43.7 | 54.4 | 58.2 | 40.9 | 51.3 | 54.8 | - | - | - |
| | CNN | 46.3 | **76.7** | **85.7** | 32.4 | 56.6 | 68.1 | 24.1 | 44.3 | 54.1 | 33.7 | 60.3 | 71.8 | - | - | - |
| | Global T. | 48.3 | **75.8** | **84.5** | 45.5 | 71.7 | 81.4 | 45.8 | 70.2 | 80.4 | 44.1 | 69.8 | 79.4 | - | - | - |

Figure 11: Models' performance averaged over 4 runs of different models for *closest string retrieval* on the Qiita dataset (1k reference and 1k query sequences, disjoint from training set).

**Results**   Figure 11 shows that convolutional and attention-based data-dependent models significantly outperform the baselines even when these operate on larger dimensions. In terms of distance functions, the cosine distance achieves performances on par with the hyperbolic. An explanation is that for a set of points on the same hypersphere, the ones with the smallest cosine or hyperbolic distance are the same. The models trained with MSE of pairwise distances and the ones with triplet loss from Section 5 performed similarly except for the hyperbolic space where the triplet loss produces unstable training. The stabilisation of the triplet loss in the hyperbolic space and further comparisons between the two training frameworks are left to future work.

# G   Steiner string approach to MSA

In this section we explain more in details the Steiner string approach to *multiple sequence alignment* introduced in Section 7.2.

**Training**   For this approach, it is necessary to train not only an encoder model but also a decoder. The resulting autoencoder is trained with pairs of sequences (and their true edit distance) which are encoded into the latent vector space and then decoded. The loss combines an edit distance approximation component and a sequence reconstruction one. The first is expressed as the MSE between the real edit distance and the vector distance between the latent embeddings. The second is expressed as the mean element-wise cross-entropy loss of the outputs with the real sequences. While this element-wise loss does not perfectly reflect the edit distance, it is an effective solution to the problem of the lack of differentiability of the latter. Therefore, given two strings $s_1$ and $s_2$ of length $n$ and a vector distance $d$, the loss of a model with encoder $f_\theta$ and decoder $g_{\theta'}$ is:

$$L(\theta, \theta') = \underbrace{(1 - \alpha)\, L_{\mathrm{ED}}(\theta)}_{\text{edit distance}} + \underbrace{\alpha\, L_{\mathrm{R}}(\theta, \theta')}_{\text{reconstruction}} \tag{12}$$

$$\text{where}\quad L_{\mathrm{ED}}(\theta) = \left(n^{-1}\, ED(s_1, s_2) - d(f_\theta(s_1), f_\theta(s_2))\right)^2$$

$$\text{and}\quad L_{\mathrm{R}}(\theta, \theta') = \frac{1}{2n} \sum_{i=0}^{n-1} \left(H(s_1[i], g_{\theta'}(f_\theta(s_1))[i]) + H(s_2[i], g_{\theta'}(f_\theta(s_2))[i])\right)$$

where $\alpha$ is a hyperparameter that controls the trade-off between the two components and $H(c, \hat{c}) = c \log(\hat{c}) + (1 - c) \log(1 - \hat{c})$ represents the cross-entropy.

One issue with this strategy is that the decoder is not learning to decode any point in the continuous space, but only those of the discrete subspace of points to which the generator maps some sequence from the domain. This creates a problem when, at test time, we try to decode points that are outside the subspace hoping to retrieve the string that maps to the point in the subspace closest to it. To alleviate this issue, during training, Gaussian noise is added to the embedded point in the latent space before decoding it, which forces the decoder to be robust to points not produced by the encoder. To make the noisy model trainable with gradient descent, we employ the reparameterization trick commonly used for Variational Auto-Encoders [74] making the randomness an input to the model. Therefore, the reconstruction loss becomes:

$$L_{\mathrm{R}}(\theta, \theta', \epsilon) = \frac{1}{2n} \sum_{i=0}^{n-1} \left(H(s_1[i], g_{\theta'}(f_\theta(s_1) + \epsilon_{1i})[i]) + H(s_2[i], g_{\theta'}(f_\theta(s_2) + \epsilon_{2i})[i])\right) \tag{13}$$

where $\forall i, j\ \ \epsilon_{ij} \sim \mathcal{N}(0, \sigma^2 \mathbb{I})$ and $\sigma$ is a hyperparameter.

In the hyperbolic space adding noise distributed with a Euclidean Gaussian distribution would not distribute uniformly, therefore we Wrapped Normal generalisation of the Gaussian distribution to the Poincaré ball [75] was used. Finally, for the cosine space, we normalise the outputs of the encoder and the input of the decoder to the unit hyper-sphere.

**Testing**   At test time, given a set of strings, we want to obtain an approximation of the Steiner string, which minimises the consensus error (sum of the distances to the strings in the set). In the sequence space with the edit distance finding the median point is a hard combinatorial optimisation problem. However, in the space of real vectors with the distance functions used in this project, it becomes a relatively simple procedure which can be done explicitly in some cases (e.g. with square distance) or using classical optimisation algorithms[5]. Therefore, the Steiner string $s^*$ of a set of strings $S$ is approximated by:

$$s^* = \arg\min_{s'} \sum_{s_i \in S} ED(s', s_i) \approx g_{\theta'}\left(\arg\min_x \sum_{s_i \in S} d(x, f_\theta(s_i))\right) \tag{14}$$

The continuous optimisation is performed using the COBYLA [76] (for the hyperbolic distance) and BFGS [77, 78, 79, 80] (for all the others) algorithms implemented in the Python library `SciPy` [81].

---

[5]If the distance function is convex such as in the Euclidean case, the resulting optimisation problem is also convex.

The produced predictions are then discretised to obtain actual sequences taking the most likely character for each element in the sequence and then evaluated by computing their average consensus error:

$$E(\hat{s}^*) = \frac{1}{|S|} \sum_{s' \in S} ED(\hat{s}^*, s') \qquad (15)$$