# OpenReview forum: "Neural Distance Embeddings for Biological Sequences"
_NeurIPS.cc/2021/Conference — NeurIPS 2021 Poster_

### Official Review · Reviewer_yqKf · 2021-07-12

**Rating:** 7
**Confidence:** 4

**Summary:**

This paper presents an encoder-based framework  (called NeuroSEED) to learn the embeddings of DNA sequences in geometric vector spaces that preserve edit distance between them. Learning an efficient representation that captures the underlying hierarchical relationships allows faster applications of downstream applications like Edit distance Approximation (EDA), Closest String Retrieval (CSR), Multiple Sequence Alignment (MSA), and Hierarchical clustering (HA). The paper explores a variety of neural network models for encoding the sequences as well as different geometric vector spaces. The tasks are formulated as unsupervised (with general MSE loss) and supervised (with tailored loss functions).  It reports hyperbolic space to give the state-of-the-art performance for all the downstream tasks and claims a faster compute time (in terms of time complexity) when using these embeddings. The results are reported on two relevant microbiome analysis datasets - Qiita and RT988.

**Limitations And Societal Impact:**

-The paper discusses limitations and potential negative impacts in the supplementary

-It would be useful for the paper to discuss NeuroSEEDs limitations (if any) when applied to new types of datasets (beyond microbiome analysis)


**Main Review:**

Strengths:

++ The paper presents an interesting idea of preserving the edit distance when encoding the genomic sequences in geometric vector spaces.

++ The problem to accurately capture the underlying biological information in terms of hierarchy and edit distance is an important one and potentially useful for the field.

++ The results are presented for 2 datasets and different settings and frameworks. The extensions to Dasgupta’s cost for HC and Steiner string approximation problem for MSA demonstrate the flexibility of NeuroSEED to task-specific formulations.

++ The paper is clear and well written


Weaknesses:

-- For a paper that claims fast approximations for downstream applications, it does not have a runtime result table in the main paper or supplementary recording the runtimes of all the baselines and the proposed framework. While some approximations of the runtime and time complexity reported do suggest faster times, it would be very useful for the reader to see and compare the run times (including training/test) when running existing methods and NeuroSEED.

-- It is important to focus on how much time does the model saves once it is trained. However,
it would also be useful for the paper to comment on how the hyperparameter tuning times would affect the overall runtime of NeuroSEED on a new task compared to running existing methods.

-- While the hyperbolic space emerges as the best performing geometric space for the embeddings, that is not the case for the encoder architecture. Can the paper recommend a default framework that would give overall good performance across datasets/tasks and still be fast?

-- Alignment-free methods like kmacs seem to allow for mismatches. Does the k-mer baseline, formulated in the paper, incorporate mismatches when approximating edit distance using k-mers? If not, could that potentially result in a lower performance than expected by using existing methods like kmacs?


**Time Spent Reviewing:**

3

---

> ### Author Response · Authors · 2021-08-10
> **Response to Reviewer yqKf**
>
> We thank the reviewer for the positive feedback. We are glad that the reviewer found the idea "interesting" and "potentially useful to the field" and the paper "clear and well written". The reviewer also highlights as weaknesses of the paper the brevity of the discussion in the paper of certain topics such as training time, hyperparameter tuning, best architectures and further baselines. We discuss all these topics below and we will improve their coverage in the paper.
>
> **Firstly, the reviewer wonders how hyperparameter tuning affects the runtime improvements.**
>
> When moving between different datasets and tasks we found that almost no hyperparameter tuning was required, as can be noticed by comparing the tuned hyperparameters provided in the READMEs of the linked repository. We expect this to hold when training on other DNA reads datasets and in general we found the framework to be not very sensitive to hyperparameter choice. We will add a discussion with more details on this topic in the Appendix.
>
> **The reviewer asks whether it is possible to make recommendations on a default architecture that would give good performance across tasks.**
>
> We found convolutional and attention-based models to perform best across almost all tasks with the former being faster. We expect CNNs to be the best choice for edit distance-like distance functions, while the attention models could be a default choice for more complex interactions between sequences. We will add this discussion to the Appendix of the paper as well.
>
> **The reviewer then raises the question of how other alignment-free methods like kmacs would perform in the tasks analysed.**
>
> As noted also by [1], kmacs and other more complex alignment-free methods provide significantly a smaller improvement over k-mers than NeuroSEED-type methods. These results were confirmed in preliminary experiments also on the datasets used in our work. We will nevertheless follow the reviewer's suggestion and add the results of these baselines to the result tables.
>
> **Finally, the reviewer suggests discussing the limitations when applied to different types of datasets.**
>
> To this end, we will integrate a new dataset from the Greengenes database [2] composed of genome sequences of highly variable length (from 1,111 to 2,368 bases). We report below some preliminary results of the edit distance approximation task on this dataset which confirm the observations presented in the paper.  Regarding sequences from different domains, as we discuss in Appendix A, it is important to choose a geometry capable of representing the relationship structure of the domain of interest. We, therefore, do not expect the hyperbolic space to be the best performing geometry for all possible domains. However, we hope to highlight, with our work, the importance of careful consideration of the embedding geometry when analysing a new domain.
>
> | Model  | Cosine | Euclidean | Hyperbolic |
> |--------|--------|-----------|------------|
> | 2-mer  |  16.17 |      7.98 |       5.08 |
> | 3-mer  |  11.21 |      5.58 |       5.13 |
> | 4-mer  |   5.93 |      3.87 |       5.16 |
> | 5-mer  |   3.60 |      3.43 |       5.18 |
> | 6-mer  |   3.15 |      3.48 |       5.19 |
> | Linear |   3.63 |      9.08 |       2.89 |
> | MLP    |   3.84 |      3.72 |       2.54 |
> | CNN    |   2.94 |      1.58 |       1.13 |
> Table: Edit distance approximation % RMSE for various models on the Greengenes dataset.
>
> The reviewer also suggests adding a result table showing all training and inference times of all the models. We welcome this suggestion and we will add such a table to the paper.
>
> [1] Zheng, W., Yang, L., Genco, R. J., Wactawski-Wende, J., Buck, M., & Sun, Y. (2019). SENSE: Siamese neural network for sequence embedding and alignment-free comparison.
>
> [2] McDonald, D., Price, M. N., Goodrich, J., Nawrocki, E. P., DeSantis, T. Z., Probst, A., Andersen, G. L., Knight, R., & Hugenholtz, P. (2012) An improved Greengenes taxonomy with explicit ranks for ecological and evolutionary analyses of bacteria and archaea.

---

> > ### Comment · Reviewer_yqKf · 2021-08-24
> > **Response to authos' comments**
> >
> > Thank you for your comments and for addressing my concerns. After reading all the reviews and responses, I am increasing my original rating.

---

### Official Review · Reviewer_HK7w · 2021-07-14

**Rating:** 7
**Confidence:** 4

**Summary:**

This paper proposes NeuroSEED to embed strings into vectors for edit distance approximation. The key insight is that hyperbolic space is more suitable for embedding the hierarchical of biological sequences. Extensive experiments are conducted to show that NeuroSEED significantly improves the efficiency of hierarchical clustering and multiple sequence alignment.



**Limitations And Societal Impact:**

Yes

**Main Review:**

I enjoyed the paper as using hyperbolic space makes sense and improving the efficiency of biological tasks is important.

However, the authors may need to tune down the contribution of NeuroSEED to properly acknowledge the contribution of existing works. In the introduction, please state explicitly that there already exist works that embed strings into vectors for edit distance approximation. I think NeuroSEED conducts a nice survey and experimental study of existing edit distance embedding methods. However, it is a bit overclaimed to say that NeuroSEED is the first to propose a general framework for edit distance embedding because the workflow of (i) using a model to map string to vector and (ii) choosing a vector space and defining a loss function for training is already utilized in existing works such as [22, 23] (maybe obvious for a machine learning pipeline).

In addition to biological tasks, the generality of hyperbolic space may need to be examined for other use cases of edit distance. [22, 23] consider string similarity search and deal with datasets containing strings with a high variation in length (while datasets in this paper contain string with similar length). In addition, for string similarity search, it is important to enlarge the distance gap between similar string pairs and dissimilar string pairs (maybe more important than accurately approximate edit distance). I am curious whether using hyperbolic space leads to better performance for string similarity search. If not, I think a more appropriate title of this paper maybe “Neural Distance Embedding for Biological Sequence”, such that the limitation of the proposed method is explicitly stated.

**Time Spent Reviewing:**

4

---

> ### Author Response · Authors · 2021-08-10
> **Response to Reviewer HK7w**
>
> We are glad that the reviewer enjoyed our work and we thank the reviewer for the positive feedback and for noting that "improving the efficiency of biological tasks is important". We also welcome the reviewer's suggestion to better clarify the contribution of existing methods from the introduction which we will include in the paper. The reviewer suggests possible extensions to examine the generality of the method to different use cases that we discuss below.
>
> **Firstly, the reviewer wonders whether the improvements provided by the hyperbolic space would hold in the case that the sequences analysed have "high variation in length".**
>
> To respond to this point we have decided to add a dataset composed of sequences from the Greengenes dataset [1] which have lengths that vary between 1,111 and 2,368 bases. Preliminary results of the edit distance approximation task on this dataset confirm the observations already reported in our work with respect to the advantages provided by NeuroSEED methods and by the hyperbolic space. These will be detailed in the next version of the paper.
>
> | Model  | Cosine | Euclidean | Hyperbolic |
> |--------|--------|-----------|------------|
> | 2-mer  |  16.17 |      7.98 |       5.08 |
> | 3-mer  |  11.21 |      5.58 |       5.13 |
> | 4-mer  |   5.93 |      3.87 |       5.16 |
> | 5-mer  |   3.60 |      3.43 |       5.18 |
> | 6-mer  |   3.15 |      3.48 |       5.19 |
> | Linear |   3.63 |      9.08 |       2.89 |
> | MLP    |   3.84 |      3.72 |       2.54 |
> | CNN    |   2.94 |      1.58 |       1.13 |
> Table: Edit distance approximation % RMSE for various models on the Greengenes dataset.
>
> **Lastly, the reviewer wonders whether the improvements hold for the task of "string similarity search" and whether they persist in different string domains.**
>
> Regarding the string similarity search (referred to as "closest string retrieval"), we provide some experiments in Appendix F on the 16S rRNA datasets showing that the geometry of the embedding space still plays an important role. Regarding different types of sequences, as we discuss in Appendix A, it is important to choose a geometry capable of representing the relationship structure of the domain of interest. We, therefore, do not expect the hyperbolic space to be the best performing geometry for all possible domains. However, we hope to highlight, with our work, the importance of careful consideration of the embedding geometry when analysing a new domain. We welcome the suggestion of the reviewer and will change the title of the paper to "Neural Distance Embeddings for Biological Sequences".
>
> [1] McDonald, D., Price, M. N., Goodrich, J., Nawrocki, E. P., DeSantis, T. Z., Probst, A., Andersen, G. L., Knight, R., and Hugenholtz, P. (2012) An improved Greengenes taxonomy with explicit ranks for ecological and evolutionary analyses of bacteria and archaea.

---

> > ### Comment · Reviewer_HK7w · 2021-08-19
> > **Comments after rebuttal**
> >
> > I have read the rebuttal and think the authors have addressed my concerns. After going over the comments of other reviewers and the author replies, I decide to keep my original rating.

---

### Official Review · Reviewer_QePG · 2021-07-16

**Rating:** 6
**Confidence:** 2

**Summary:**

The paper tackles the problem of approximating edit distances between biological sequences. For this purpose, the authors propose an encoder that maps sequences to a vector space, and is trained in a way such that the distance between encoded representations matches the edit distance between underlying sequences. The authors show multiple experiments with different architectures (Linear, Transformer, CNN, GRU, etc.) and different geometries (cosine, euclidean, hyperbolic, etc.). Overall, the approach is able to approximate well the edit distance while being much faster to compute.

**Limitations And Societal Impact:**

Yes

**Main Review:**

Disclaimer: I am not very knowledgeable about biological sequences.
I found the paper interesting, well-written and easy to follow. The idea of using neural networks to capture the statistics of the distribution in an embedding space seems to work well and to avoid doing M^2 operations to compute pairwise distances. I think it is an interesting new use case for neural networks, where it seems to work well. The extension to hierarchical clustering and multiple sequence alignment are interesting and seem to perform well.

Questions:
- with what architecture was the running time computed ? Not all proposed models have a linear complexity with respect to the sequence length
- why is there a decoder in Figure 1 that is not mentionned in the caption and does not seem to be used at all?


**Time Spent Reviewing:**

2

---

> ### Author Response · Authors · 2021-08-10
> **Response to Reviewer QePG**
>
> We thank the reviewer for the time taken to review our work. We are glad that the reviewer found "the paper interesting and well-written" and was persuaded by the ideas and the results presented. Below we answer the two questions that were raised and we will clarify these points in the paper.
>
> **Q: with what architecture was the running time computed? Not all proposed models have a linear complexity with respect to the sequence length**
>
> A: The times reported for the NeuroSEED were either the upper bound (maximum between the different models) or the full range (minimum-maximum). To avoid this confusion we will add a table with both training and inference times for all the models to the final version of the paper. The time taken for each model varies depending on the architecture and to a lesser extent the distance function used. This difference is, however, small when compared to the improvement over the baselines. Assuming constant embedding size, the global transformer is the only model with non-linear time complexity, this limitation enables accurate modelling of complex mutations such as inversions and duplications and is alleviated by the high levels of parallelism possible on GPUs.
>
> **Q: why is there a decoder in Figure 1 that is not mentioned in the caption and does not seem to be used at all?**
>
> A: The decoder shown in Figure 1 is used in Section 7.2, we will add a reference in the caption to clarify this point. We included the decoder in the framework because we believe it is potentially very useful in many downstream applications. The Steiner string MSA task (Section 7.2) is provided as evidence that such decoding is possible.

---

> > ### Comment · Reviewer_QePG · 2021-08-30
> > **Thanks to the authors for their response**
> >
> > I thank the authors for their response.
> > Given their response as well as other reviews, I choose to maintain my score. I think the paper provides a valuable contribution, and that there is amble experimental support provided in the paper to the proposed method.

---

### Official Review · Reviewer_Pdgk · 2021-07-16

**Rating:** 4
**Confidence:** 4

**Summary:**

This paper offers two contributions - the first is a formalization of a general encoder-decoder framework for learning representations of biological sequences, where the distance in the representation space corresponds to the edit distance between the sequences (NeuroSEED). The second is the use of hyperbolic geometry to embed the sequences. The authors then demonstrate the efficacy and runtime improvements of their method on a variety of tasks, including approximating edit distance, MSA and hierarchical clustering.

**Ethical Concerns:**

No ethical concerns.

**Limitations And Societal Impact:**

There is no discussion of limitations and societal impact in the main text, but some discussion of limitations is provided in the appendix. I personally do not find the lack of societal impacts for this work to be an issue.

**Main Review:**

Overall, the manuscript is well written and the work is solid, but the exact training method and models compared are not clear from the main text. The significance and practicality of the method seem low, given that a model needs to be trained for each dataset rather than being pre-trained. Furthermore, the novelty is weak and discussion and comparison with other frameworks is missing. The elephant in the room here being locality sensitive hashing-based methods. Specific comments follow below.

1.	The idea to use an encoder-decoder framework to embed sequences in a latent space is not new, the application of such a framework for biological sequences is not new  (see e.g., https://www.nature.com/articles/s41598-018-34533-1, https://link.springer.com/article/10.1186/s12859-017-1700-2, http://proceedings.mlr.press/v70/mueller17a.html), and the use of hyperbolic geometry as the embedding geometry is not new. However, the specific combination of these methods in application to biological sequences is, to my knowledge, novel.
2.	The framework is only evaluated on two 16S rRNA datasets rather than whole genome sequencing or other more diverse sequencing datasets. Furthermore, the models are trained on each dataset specifically. This seems like a major practical limitation. How does this pre-training time factor into a sequence analysis project? The reported training time is long enough that it seems to remove any time benefits downstream. It seems somewhat misleading to focus on downstream clustering or pairwise distance calculation times when the model needs to be trained first and typically analysis is only performed once for a given dataset leading to limited amortization. On top of which, training time should increase with larger datasets. Is it possible to pre-train these models and have them generalize well? Do these models work on more diverse sequence datasets? What about sequences with larger disparities in length?
3.	Section 7.1 is unclear in explaining the approach - in particular line 241 - it is unclear how the linear layer maps sequences of different lengths to a hyperbolic space, or how the model is trained. Furthermore, this is referred to as an encoder-decoder framework, but a decoder is only used in one experiment and the details of its structure are fuzzy.
4.	Additionally, the tables often include methods first established in other papers - it is unclear whether the entries in these tables are 1) original methods and original experiments by the authors, 2) methods taken from other papers but new experiments done by the authors, or 3) methods and experiments taken from other papers.
5.	No comparison with locality sensitive hashing-based methods or even discussion of these methods in biological sequence analysis and other heuristic methods for sequence clustering that are widely used (e.g., https://www.nature.com/articles/s41467-018-04964-5, https://academic.oup.com/bioinformatics/article/22/13/1658/194225, https://www.nature.com/articles/nbt.3988). Even BLAST uses heuristics to find similar sequences before alignment (https://pubmed.ncbi.nlm.nih.gov/2231712/) which is not mentioned.
6.	This work considers edit distance, but in practice biological sequence comparison is not performed based on edit distance but rather substitution scores between amino acids. These are not distance functions. How does this fit into the proposed framework?
7.	“Again, the results reported in Figure 6 show that the alignment scores obtained when using the NeuroSEED heuristics with attention models are not statistically different from those obtained with the ground truth distances.” – line 224. Reporting these results as percentage differences from CLUSTAL using the ground truth edit distances as the guide tree is confusing. It would be nice to report the scores directly. How does this compare to the results obtained by running CLUSTAL with default settings?

I think the significance of this work is limited. It’s hard to call the formalization of the framework a useful contribution, because the idea of encoder-decoder-loss-geometry is already a standardized concept in the field. Furthermore, the offered performance and speedup benefits are limited by the need to train the model on each dataset and the generalizability to larger and more diverse sequence datasets is unknown.


**Time Spent Reviewing:**

3

---

> ### Author Response · Authors · 2021-08-10
> **Response to Reviewer Pdgk**
>
> We thank the reviewer for the thorough review. The reviewer seemed to appreciate "the manuscript well written and the work solid", but raised some concerns especially with respect to the training time and comparison to locality sensitive hashing. In our response, we highlight how, considering the scale of the datasets and applications of the biological sequences analysed, the training time is relatively small and does not limit its applicability. Then, we argue that LSH methods should be considered as complementary not substitutive of our framework. Below we also respond to all the questions raised by the reviewer and we will integrate them in the final version of the paper.
>
> **Firstly the reviewer raises the question of "does this pre-training time factor into a sequence analysis project?" suggesting that this would "remove any time benefits downstream".**
>
> The pre-training was done on 5-7k sequences and takes a time similar to doing the pairwise alignment of those sequences (0.5-3h). However, microbiome studies (such as those from which the sequences were taken) typically analyse millions of sequences to which the trained model can 'extrapolate' as demonstrated in the experiments (the training time does not scale with data used for inference). On such large amounts of sequences, traditional alignment methods are hopeless, while our method can provide good approximations. Moreover, in many clinical applications of microbiome analysis, fast inference speed is critical. In the case of infections where microbiome analysis is used to decide the therapy, studies [1] show that every hour of effective delay in treatment increases mortality by ~8%, therefore, having a pretrained model (for the type of sequences analysed but not necessarily from the same individual) could provide a critical speedup. Finally, the method proposed in Section 7.1 does not require any pre-training but still gives significant speedups over the baselines. Therefore we believe that pre-training time is relatively small and does not have a significant impact on the strong downstream improvements provided by NeuroSEED. We will clarify this point in the paper.
>
> **The reviewer also raises the important question of how would the "models work on more diverse sequence datasets" and with "sequences with larger disparities in length".**
>
> To the first point, we believe the datasets we considered already provide evidence as they are composed of sequences from the highly variable regions of the 16S rRNA gene, reaching pairwise dissimilarity levels close to those expected in random independent strings. To respond to the second point we have added a dataset composed of sequences from the Greengenes dataset [6] which have lengths that vary between 1,111 and 2,368 bases. Preliminary results of the edit distance approximation task on this dataset (summarised in the table below) confirm the observations already reported in our work confirming that length disparities can be handled by the proposed framework. These experiments will be detailed in the next version of the paper.
>
> | Model  | Cosine | Euclidean | Hyperbolic |
> |--------|--------|-----------|------------|
> | 2-mer  |  16.17 |      7.98 |       5.08 |
> | 3-mer  |  11.21 |      5.58 |       5.13 |
> | 4-mer  |   5.93 |      3.87 |       5.16 |
> | 5-mer  |   3.60 |      3.43 |       5.18 |
> | 6-mer  |   3.15 |      3.48 |       5.19 |
> | Linear |   3.63 |      9.08 |       2.89 |
> | MLP    |   3.84 |      3.72 |       2.54 |
> | CNN    |   2.94 |      1.58 |       1.13 |
> Table: Edit distance approximation % RMSE for various models on the Greengenes dataset.
>
> **Another point raised by the reviewer is the lack of comparison with locality-sensitive hashing based methods.**
>
> We will argue that while surely relevant, these are not directly comparable to our method. The LSH methods referenced by the reviewer tackle the tasks of sequence clustering [4,5] and local alignment [7] which are not part of the focus of our work (which concentrates on edit distance approximation and its applications to HC and MSA). Moreover, they use as subroutines embeddings or heuristics based on alignment-based or alignment-free methods such as k-mers analysed in our work, and, therefore, fall into the limitations we highlighted in the paper. We instead see LSH methods as complementary to our work. As already mentioned in the paper and in the supplementary material, clustering and closest string retrieval can be speeded up using traditional LSH methods on top of the NeuroSEED embeddings. An alternative approach would be to substitute the alignment-free or alignment-based edit distance approximation steps in methods such as [4,5] with those we propose, this is a similar approach to the one for MSA presented in Section 6. We nevertheless thank the reviewer for raising this point and we will further discuss the relationship between NeuroSEED and LSH in the paper.
>
> The reviewer further asked a series of clarification questions that we respond to below:
>
> **Q: Section 7.1 is unclear in explaining the approach, it is unclear how the linear layer maps sequences of different lengths to a hyperbolic space, or how the model is trained.**
>
> A: We thank the reviewer for raising this point and we will try to clarify this section. As in the previous section, the linear layer takes as input a one-hot representation of the sequences padded to the maximum length in the dataset and outputs a vector of a specific length. The training is done using gradient descent with the relaxed Dasgupta's cost as loss function:
>
> $$C(Z; w,\tau)=\sum_{ijk}(w_{ij}+w_{ik}+w_{jk}-w_{ijk}(Z; w,\tau))+2\sum_{ij}w_{ij}$$
> $$ \text{where}\ w_{ijk}(Z; w,\tau)=(w_{ij}, w_{ik}, w_{jk})\ \cdot\ \sigma_\tau(d_o( z_i \vee z_j), d_o( z_i \vee z_k),d_o(z_j \vee z_k))^\top,$$
> $Z$ are the leaves, $w_{ij}$ the similarity, $i\vee j$ the hyperbolic LCA, $d_o$ the distance from the origin, $\sigma_\tau(\cdot)$ the scaled softmax and i, j and k are triplets for which distances are available. More details on this can be found in [2].
>
> **Q: This is referred to as an encoder-decoder framework, but a decoder is only used in one experiment and the details of its structure are fuzzy.**
>
> A: We included the decoder in the framework because we believe it is potentially very useful in many downstream applications. The Steiner string MSA task is provided as evidence that such decoding is possible. As typical in the literature, the structure of the decoder is very similar to the one of the encoders reversed: a fixed vector as input and outputs a tensor of the same length of the padded input sequences and softmax applied to each element. We also provide the code and data used for every experiment for the readers interested in reproducing/extending our work.
>
> **Q: The tables often include methods first established in other papers - it is unclear whether the entries in these tables are 1) original methods and original experiments by the authors, 2) methods taken from other papers but new experiments done by the authors, or 3) methods and experiments taken from other papers.**
>
> A: Unfortunately the previous works [8,9] we cited do not include publicly available code and/or data used (even after explicit requests to the authors). Therefore, all the experiments were conducted by us. As we tried to clarify in Table 1 some of the results reported in the tables correspond very closely (in some aspects) to previous works. We will further highlight these similarities and differences adding explicit links in the tables and their captions.
>
> **Q: This work considers edit distance, but in practice biological sequence comparison is not performed based on edit distance but rather substitution scores between amino acids. These are not distance functions. How does this fit into the proposed framework?**
>
> A: The distance function is taken as a black box by the models therefore 'more complex' substitution matrices or gap penalties can be used. To obtain good performances the function used on the sequences should have similar characteristics to the one used in the embedding space (for example, with respect to being non-negative or symmetric). For amino acids, for example, we expect one of the different metric variants of the classical substitution matrices such as mPAM250 [3] to be best suited to be used in combination with the hyperbolic distance. We will clarify this point in the paper.
>
> **Q: Reporting these results as percentage differences from CLUSTAL using the ground truth edit distances as the guide tree is confusing. It would be nice to report the scores directly. How does this compare to the results obtained by running CLUSTAL with default settings?**
>
> A: We thank the reviewer for the suggestion, we will add the raw scores to the table as well, they were removed because they are not easily interpretable (all around -10000000). The difference with CLUSTAL's default settings is the different scoring of the similarity, therefore the scores obtained are not directly comparable.
>
> We hope that our answers have clarified the points raised and increased the reviewer's confidence in our work.
>
> [1] Kollef et al. Broad-spectrum antimicrobials and the treatment of serious bacterial infections: getting it right up front.
>
> [2] Chami et al. From trees to continuous embeddings and back: Hyperbolic hierarchical clustering.
>
> [3] Xu et al. A metric model of amino acid substitution.
>
> [4] Li et al. Cd-hit: a fast program for clustering and comparing large sets of protein or nucleotide sequences.
>
> [5] Steinegger et al. Clustering huge protein sequence sets in linear time.
>
> [6] McDonald et al. An improved Greengenes taxonomy with explicit ranks for ecological and evolutionary analyses of bacteria and archaea.
>
> [7] Altschul et al. Basic local alignment search tool.
>
> [8] Zheng et al. SENSE: Siamese neural network for sequence embedding and alignment-free comparison.
>
> [9] Chen et al. Predicting Alignment Distances via Continuous Sequence Matching.

---

### Author Response · Authors · 2021-08-10
**Summary of responses and changes**

We thank all the reviewers for the time they spent reviewing our work. We are glad that all the reviewers appreciated our work and found the paper well written. We did our best to respond to all individual questions from the reviewers in the direct response and we will try to integrate every piece of feedback in the final version of the paper. Below we summarise some of the main changes that we made to respond to the reviewers' feedback:

1. We have added an additional dataset using sequences from the Greengenes database [1]. This contains DNA sequences that are significantly longer than the ones from the other datasets and with high variation in length (1,111 and 2,368 bases). The table below shows the preliminary results of the edit distance approximation task on this dataset which confirm all the observations presented in the paper. We will provide full results on the dataset and further details on the dataset and training in the paper.

| Model  | Cosine | Euclidean | Hyperbolic |
|--------|--------|-----------|------------|
| 2-mer  |  16.17 |      7.98 |       5.08 |
| 3-mer  |  11.21 |      5.58 |       5.13 |
| 4-mer  |   5.93 |      3.87 |       5.16 |
| 5-mer  |   3.60 |      3.43 |       5.18 |
| 6-mer  |   3.15 |      3.48 |       5.19 |
| Linear |   3.63 |      9.08 |       2.89 |
| MLP    |   3.84 |      3.72 |       2.54 |
| CNN    |   2.94 |      1.58 |       1.13 |
Table: Edit distance approximation % RMSE for various models on the GreenGenes dataset.

2. We added a table showing the training and inference times for each model and further discussed the implications of the improvements in inference time shown by NeuroSEED. For example, in the case of infections where microbiome analysis is used to decide the therapy, studies [1] show that every hour of effective delay in treatment increases mortality by ~8%, therefore, having a pretrained model (for the type of sequences analysed but not necessarily from the same individual) could provide a critical speedup.

3. We followed the reviewers' suggestion to further discuss the relationship between NeuroSEED and other types of models such as alignment-free methods like kmacs and locality sensitive hashing. Where possible we also added direct performance comparisons.

[1] McDonald, D., Price, M. N., Goodrich, J., Nawrocki, E. P., DeSantis, T. Z., Probst, A., Andersen, G. L., Knight, R., and Hugenholtz, P. (2012) An improved Greengenes taxonomy with explicit ranks for ecological and evolutionary analyses of bacteria and archaea.

---

### Decision · Program_Chairs · 2021-09-27

**Decision:**

Accept (Poster)

**Comment:**

The paper studies methods for embedding sequences into geometric spaces that approximately preserve the edit distance. The key finding is that embedding into hyperbolic spaces yields (significantly) lower distortion than embedding into other geometric spaces. The finding is supported by experiments on multiple data sets, in the context of several downstream applications: similarity search, clustering and multiple alignment.

Despite a few rounds of discussion, the reviewers did not reach a consensus. Most of the reviews are positive, with some reviewers being very positive. At the same time though, the negative review raises valid points, including (1) the basic edit distance (where all operations have unit cost) does not occur often in applications, so the practical impact of the proposed method is unclear (2) in the context of similarity search, there is no comparison to indexing methods that are faster than linear scan (e.g., LSH); given that fast algorithms for hyperbolic spaces are not well developed, it could be that the more accurate embeddings into hyperbolic spaces might actually yield less efficient algorithms than the less accurate embeddings into (say) the Euclidean space.

That said, the paper does introduce an interesting idea, and it is likely that its findings will stimulate further research on embeddings into (and algorithms for) hyperbolic spaces. So, I recommend accepting the paper,  even though at present its practical impact is not guaranteed.